

# Trans-Arctic vicariance in *Strongylocentrotus* sea urchins

Jason A. Addison[1] and Jinhong Kim[1,2]

[1] Biology, University of New Brunswick, Fredericton, New Brunswick, Canada
[2] Department of Biochemistry and Molecular Biology, Dalhousie University, Halifax, Nova Scotia, Canada

Corresponding author
Jason A. Addison, jaddison@unb.ca

## ABSTRACT

The sea urchins *Strongylocentotus pallidus* and *S. droebachiensis* first invaded the Atlantic Ocean from the Pacific following the opening of the Bering seaway in the late Miocene. While trans-Arctic dispersal during the Pleistocene is thought to have maintained species' integrity, a recent genomic analysis identified a reproductively isolated cryptic species within *S. droebachiensis*. Based on previous studies, the distribution of one of these lineages (*S. droebachiensis* W) includes the shallow water habitats of the northwest Atlantic and Pacific, while the other (*S. droebachiensis* E) is found throughout the shallow habitat in the northeast but is mostly restricted to deep habitats (>65 m) in the northwest Atlantic. However, since genetic variation within *S. droebachiensis* has been largely unstudied in the north Pacific and Arctic oceans, the biogeography of the cryptic species is not well known, and it is difficult to identify the mechanisms driving population subdivision and speciation. Here we use population genetic analyses to characterize the distribution of each species, and to test hypotheses about the role of vicariance in the evolution of systematic and genomic divergence within the genus. We collected individuals of all three *Strongylocentrotus* species ($n = 365$) from 10 previously unsampled locations in the northeast Pacific and north Atlantic (Labrador Sea and Norway), and generated mtDNA sequence data for a 418 bp fragment of cytochrome *c* oxidase subunit I (*COI*). To assess the biogeography of all three species, we combined our alignment with five previously published data sets (total $n = 789$) and used statistical parsimony and maximum likelihood to identify species and characterize their distribution within and among oceans. Patterns of haplotype sharing, pairwise $F_{ST}$, and hierarchical analyses of molecular variance (AMOVA) identified trans-Arctic dispersal in *S. pallidus* and *S. droebachiensis* W, but other than 5 previously reported singletons we failed to detect additional mtDNA haplotypes of *S. droebachiensis* E in the north Pacific. Within the Atlantic, patterns of habitat segregation suggests that temperature may play a role in limiting the distribution of *S. droebachiensis* E, particularly throughout the warmer coastal waters along the coast of Nova Scotia. Our results are consistent with the cycles of trans-Arctic dispersal and vicariance in *S. pallidus* and *S. droebachiensis* W, but we suggest that the evolution of Atlantic populations of *S. droebachiensis* E has been driven by persistent trans-Arctic vicariance that may date to the initial invasion in the late Pliocene.

## INTRODUCTION

The global biogeography of Strongylocentrotid sea urchins was shaped by the trans-Arctic interchange following the initial opening of the Bering seaway in the late Miocene (5.5–5.0 Mya; *Marincovich & Gladenkov, 2001*; *Gladenkov et al., 2002*), and fossil evidence reveals that *Strongylocentrotus droebachiensis* (along with many other Pacific taxa) reached western Europe by the late Pliocene (*Durham & MacNeil, 1967*). Following the initial invasion, eustatic sea level changes during the Pleistocene ice ages (2.4–0.2 Mya) periodically restricted dispersal across the Arctic Basin, causing widespread isolation and vicariance in the north Atlantic (*Hewitt, 1996*; *Cunningham & Collins, 1998*). As a result of these processes, molecular evidence from trans-Arctic taxa indicates a complex pattern of inter- and intra-specific divergence, with species positioned along a continuum between complete reproductive isolation and panmictic populations (see *Laakkonen et al., 2021*). There has been consensus that both genetic diversity and species integrity have been maintained among populations of *Strongylocentrotus* sea urchins from the Pacific (ancestral) and Atlantic (colonized) as a result of gene flow across the Arctic between 0.40–0.11 Mya (*Palumbi & Kessing, 1991*; *Palumbi & Wilson, 1990*; *Addison & Hart, 2005*; *Laakkonen et al., 2021*). However, the recent discovery of a cryptic species within the north Atlantic population of *S. droebachiensis* (*Addison & Kim, 2018*) suggests a more complicated role of vicariance in the evolution of the genus, demanding a re-evaluation of both the biogeography and population genetics throughout the range.

Like many marine invertebrates, *Strongylocentrotus* sea urchins are broadcast spawners with long-lived planktonic larvae (4 to 21 weeks; *Strathmann, 1978*) capable of high rates of dispersal with gene flow. Genetic studies of *S. droebachiensis* (*Müller, 1776*) in the north Atlantic detected local panmixis in the northwest (*Addison & Hart, 2004*; *Addison & Hart, 2005*), a small but significant latitudinal gradient in the northeast (Nordberg et al., 2016), and significant population substructure between the east and west coasts (*Addison & Hart, 2004*; *Addison & Hart, 2005*; *Harper, Addison & Hart, 2007*). Patterns of genetic variation at both microsatellites and mtDNA reveal lower levels of diversity in the eastern populations compared to the west, and pairwise $F_{ST}$ suggests that northwest populations are genetically more similar to the Pacific than to those from the northeast Atlantic (*Addison & Hart, 2004*; *Addison & Hart, 2005*). A more detailed study of *S. droebachiensis* in the northwest Atlantic indicates strong habitat segregation and reproductive isolation between distinct east and west mtDNA haplogroups, where the vast majority shallow-water coastal samples (<30 m) were a subset of haplotypes shared between the Pacific and the northwest Atlantic, and the offshore deep-water (>65 m) samples were identical to (or clustered with) populations from Norway and Iceland (*Addison & Kim, 2018*). These genetic patterns broadly correspond to variation in sperm morphology reported between sea urchins from the northeast Atlantic and those from the northwest Atlantic and Pacific (*Manier & Palumbi, 2008*; *Marks et al., 2008*). Together, these patterns of genetic and morphological divergence indicate that trans-Atlantic variation reflects species level differences and are not the result of limited gene flow and genetic drift between allopatric populations. Thus, there is strong evidence that the north Atlantic harbours two reproductively isolated species of *Strongylocentrotus*

sea urchins: *S. droebachiensis* W whose distribution includes the shallow water habitats of the northwest Atlantic and Pacific, and *S. droebachiensis* E that is distributed throughout the shallow habitat in the northeast and deep habitats (>65 m) at lower latitudes in the northwest Atlantic.

Cycles of trans-Arctic dispersal and vicariance have played an important role in the evolution of new species in a variety of Pacific and Atlantic taxa, including molluscs, crustaceans, echinoderms, polychaetes, fishes, mammals, and algae (*e.g.*, *Wares, 2001*; *Carr et al., 2011*; *Layton, Corstorphine & Hebert, 2016*; *Neiva et al., 2018*; *Bringloe, Verbruggen & Saunders, 2020*; *Laakkonen et al., 2021*). The surprising discovery of two independent lineages of *S. droebachiensis* suggests that allopatric speciation in this genus may have also followed the initial trans-Arctic invasion. While vicariance during the Pliocene-Early Pleistocene resulted in the speciation of the sea stars *Asterias forbesi* and *A. rubens* in the Atlantic (*Wares, 2001*), more recent vicariant histories during the Middle Pleistocene (1–0.2 Mya) have resulted in reciprocal monophyly and divergence among populations of *Solaster endeca, Pteraster militanus*, and *Crosster papposus* (*Layton, Corstorphine & Hebert, 2016*). Interoceanic divergence between allopatric Pacific and Atlantic lineages of these species ranges from 1.24% to 2.98%, and although the taxonomic status in these groups is unknown, these differences are comparable to species level differences at cytochrome *c* oxidase subunit I sequences (*COI*) among other echinoids (Echinometra 2–3%, *Palumbi et al., 1997*; Leptasterias 0.4–2.2%, *Hrincevich, Rocha-Olivares & Foltz, 2000*; Patiriella 1.1–4.3%, *Hart, Byrne & Johnson, 2003*), including the cryptic lineages of *S. droebachiensis* (2.3%, *Addison & Kim, 2018*). However, the distribution and extent of the ecological segregation of both lineages of *S. droebachiensis* throughout the north Atlantic are not well known, particularly in the Labrador Sea, where the west flowing Greenland Current is expected to connect populations across the north Atlantic (*Knutsen et al., 2007*; *Bringloe, Verbruggen & Saunders, 2020*). Furthermore, genetic variation within *S. droebacheisis* has been largely unsampled in the north Pacific, making it difficult to assess the role of trans-Arctic vicariance to patterns of evolution within the genus.

Here we extend analyses of biogeography and population genetic structure in circumpolar *Stronglyocentrotus* sea urchins to better understand the roles that trans-Arctic and trans-Atlantic dispersal have played in the systematic and genetic divergence within the genus. We aim to establish a more complete understanding of the range limits of each species by compiling previous surveys of mtDNA sequence variation and including additional sample sites for *S. pallidus* and *S. droebachiensis* throughout the Pacific and north Atlantic. *Strongylocentrotus pallidus* is a circumpolar species that is abundant in shallow water (<15 m) in the north, and deeper waters of up to 1600 m at lower latitudes (*Jensen, 1974*; *Strathmann, 1981*; *Gagnon & Gilkinson, 1994*; *Bluhm, Piepenburg & Von Juterzenka, 1998*). The current known distribution of *S. droebachiensis* W includes the northeast Pacific and the shallow water habitat of the northwest Atlantic, and *S. droebachiensis* E is the only green sea urchin found in the northeast Atlantic and in the deep offshore habitat in the northwest Atlantic. However, there is less certainty about the full distribution *of S. droebachiensis* E, because although it appears to be circumpolar, it was only detected at low frequency in the Pacific (5/29 samples; *Addison & Hart, 2005*), and none of these

haplotypes were shared with the Atlantic suggesting a lack of recent trans-Arctic dispersal. Although mtDNA from this lineage was extremely rare at lower latitudes in the shallow northwest Atlantic habitat (6/161 samples; *Addison & Hart, 2005*; *Addison & Kim, 2018*), a more complete understanding of the biogeography within the genus requires additional sampling at higher latitudes in both the northwest Atlantic and northeast Pacific. Late glacial and post-glacial trans-Arctic dispersal between populations of *S. pallidus* and *S. droebachiensis* W has resulted in both shallow inter-ocean divergence and widespread sharing of haplotypes (*Palumbi & Kessing, 1991*; *Palumbi & Wilson, 1990*; *Addison & Hart, 2005*; *Laakkonen et al., 2021*). Since the coastal habitat at high latitudes in the Pacific and northwest Atlantic is qualitatively similar to the northeast Atlantic where *S. droebachiensis* E dominates (*e.g.*, cool water, kelp; *Payne et al., 2012*; *Government of Canada, 2014*; *Bringloe & Saunders, 2019*), we expect concordant biogeography and patterns of genetic diversity in *S. droebachiensis* E if it shared a similar history of invasion, vicariance, and secondary contact. However, an absence of haplotype sharing or discordant biogeography would suggest a lack of trans-Arctic dispersal (persistent vicariance), raising the possibility that *S. droebachiensis* E diverged in allopatry following the initial trans-Arctic invasion in the late Pliocene. This study will provide insight into the mechanisms driving reproductive isolation in the northern lineages of *Strongylocentrotus* sea urchins by defining the biogeographical distribution of allopatric and sympatric populations, and quantifying genetic subdivision both within and between oceans.

## MATERIALS & METHODS

### Sampling, DNA extraction, amplification, and sequencing

To examine the range distributions of all three lineages of *Strongylocentrotus* sea urchins we compiled *COI* sequence data from previous studies (GenBank accession Nos. AY504479–AY504511, *Addison & Hart, 2005*; EF108346–EF108365, *Harper, Addison & Hart, 2007*; MG098337–MG098440, *Addison & Kim, 2018*; MT736172–MT736220, *Laakkonen et al., 2021*), and collected new samples from 10 locations throughout the north Atlantic and northeast Pacific oceans (Fig. 1; Table 1). New samples were collected from most sites at depths of 2–8 m using SCUBA, except in the Bay of Fundy (14–90 m) where samples were collected using a fixed gear dredge, and Owl's Head Nova Scotia (60 m) where collections were made using baited lobster traps as described in *Filbee-Dexter & Scheibling (2014)*. Sea urchins were collected with permission under Section 52 permits (Department of Fisheries and Oceans Canada: 356132, NL-2619-14, TNMP-2014-16478, and S-14/15-1053-NU), and the State of Alaska Department of Fish and Game (CF-17-004). Sea urchins from the northeast Atlantic at Kongsfjord and OsloFijord (Norway) were a subset of those analysed by *Norderhaug et al. (2016)* for which we generated new mtDNA sequence data. We generally observed the colour characteristics of the specimens (*i.e.*, test, tube feet, and aboral spines) described in *Jensen (1974)* and *Gagnon & Gilkinson (1994)*, but since the structures used to distinguish the species are challenging to observe under field conditions (*e.g.*, pedicellariae, spine wedges, and pore pairs), designations were ultimately made using DNA sequence data (see *Addison & Kim, 2018*). We preserved gonad tissue and/or tube

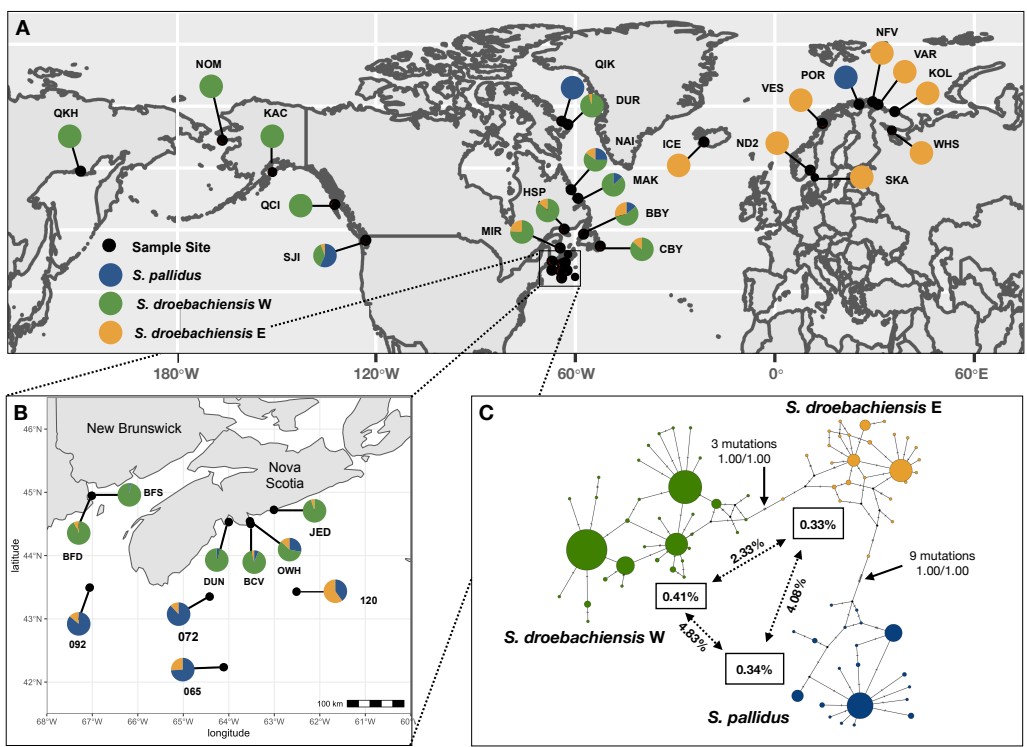

**Figure 1** (A) Sample sites of Strongylocentrotus sea urchins throughout the North Pacific and north Atlantic oceans (see Table 1 for abbreviations). Pie charts represent the proportion of mtDNA haplotypes (418 bp COI) belonging to each of the three lineages. (B) Inset map of samples collected throughout Atlantic Canada. (C) TCS haplotype network of COI mtDNA sequences from all three lineages of Strongylocentrotus sea urchins ($n = 789$) included in this study. Circle area is proportionate to the number of haplotypes sequenced and the colours of each lineage match the pie charts from A and B. Node support indicated by nonparametric bootstrap (1,000 replicates) and Bayesian posterior probability, respectively. Overall mean K2P distances are within each lineage is indicated in the boxes, and mean pairwise distances are indicated along the vectors.

feet in 95% ethanol, and extracted total genomic DNA using DNAeasy Blood and Tissue columns (QIAGEN) following the manufacturer's recommended protocols.

We targeted a fragment of the cytochrome oxidase subunit I (*COI*) mitochondrial gene using the polymerase chain reaction (PCR) primers COIJ and COIC (*Edmands, Moberg & Burton, 1996*). Following *Addison & Kim (2018)*, we performed amplifications in a 30µL volume consisting of ~4ng DNA, 1 × ThermoPol reaction buffer (New England Biolabs, NEB), 0.2 mmol dNTPs (NEB), 2.0 mmol MgSO$_4$, 0.5 µmol forward and reverse primers, and 1.0 unit of Taq polymerase (NEB). Thermal cycling conditions were 95 °C for 3 min, followed by 39 cycles of 95 °C (30s), 45 °C (30s), 72 °C (60s), and a final extension at 72 °C for 3 min. We checked amplicons using agarose gel electrophoresis and visualized with SYBR™ Safe (Invitrogen™) under UV light. Sanger sequencing using forward, reverse, or both PCR primers was conducted at the Genome Quebec Innovation Centre (McGill University, Montreal, Quebec, Canada). Sequences were edited, aligned, and trimmed to a

**Table 1  Sample location, site codes, sample size, sampling depth, and data sources for *Strongylocentrotus* sea urchins used in this study.**

| Location | Abbreviation | Sample Size (N) | depth (m) | Latitude | Longitude | COI Data Source |
|---|---|---|---|---|---|---|
| Sea of Okhotsk, Russia | OKH | 16 | n/a | 59.49400 | −150.91500 | *Laakkonen et al. (2021)* |
| San Juan Islands, WA | *SJI* | 39 | 50 | 48.33000 | −123.01000 | *Addison & Hart (2005)*; This study |
| San Juan Islands, WA | *SJI* | 40 | 30 | 48.32800 | −123.06000 | *Harper, Addison & Hart (2007)*; This study |
| *Langara Island, BC* | *LAN* | 20 | 2-8 | 54.19300 | −132.97200 | This study |
| *Massett, BC* | *MAS* | 18 | 2-8 | 54.10700 | −132.36600 | This study |
| *Kachemak Bay, Alaska* | *KAC* | 2 | 5 | 59.48500 | −151.64600 | *Laakkonen et al. (2021)* |
| Nome, AK | *NOM* | 26 | 2-8 | 64.48737 | −166.19298 | This study |
| Qikiqtarjuaq, NU | *QIK* | 5 | 2-8 | 67.56800 | −64.06700 | This study |
| Durban Island, NU | *DUR* | 19 | 2-8 | 67.03800 | −62.24900 | This study |
| Makkovik, NL | *MAK* | 31 | 2-8 | 55.10200 | −59.18000 | This study |
| Nain, NL | *NAI* | 65 | 2-8 | 56.50400 | −61.26300 | This study |
| *Bonne Bay, NL* | *BBY* | 7 | 3-15 | 49.31000 | −57.53000 | *Addison & Hart (2005)* |
| *Conception Bay, NL* | *CBY* | 7 | 3-15 | 47.38000 | −52.50000 | *Addison & Hart (2005)* |
| *Harve Saint Pierre, QC* | *HSP* | 7 | 3-15 | 50.14000 | −63.36000 | *Addison & Hart (2005)* |
| *Miramichi, NB* | *MIR* | 21 | 3-15 | 47.08000 | −64.58000 | *Addison & Hart (2005)* |
| Jeddore, NS | *JED* | 42 | 3-15 | 44.73000 | −63.01102 | *Addison & Hart (2005)* |
| Bear Cove, NS | *BCV* | 48 | 3-15 | 44.53670 | −63.54195 | *Addison & Hart (2005)* |
| Duncan Cove, NS | *DUN* | 48 | 2-30 | 44.49794 | −63.51038 | *Addison & Kim (2018)* |
| Owl's Head, NS | *OWH* | 69 | 60 | 44.52090 | −64.00069 | This study |
| *NS offshore (65m)* | *065* | 8 | 65 | 42.23480 | −64.12820 | *Addison & Kim (2018)* |
| *NS offshore (72m)* | *072* | 19 | 72 | 43.35880 | −64.42790 | *Addison & Kim (2018)* |
| *NS offshore (92m)* | *092* | 14 | 90 | 43.48620 | −67.07020 | *Addison & Kim (2018)* |
| *NS offshore (120m)* | *120* | 15 | 120 | 43.43670 | −62.50380 | *Addison & Kim (2018)* |
| *Bay of Fundy (shallow)* | *BFS* | 63 | 14 | 44.95233 | −67.01451 | This study |
| *Bay of Fundy (deep)* | *BFD* | 28 | 70-90 | 44.58000 | −67.00000 | This study |
| Hvalfjordur, Iceland | *ICE* | 12 | 10 | 64.21000 | −21.29000 | *Addison & Hart (2005)* |
| Oslo fjord, Norway | *ND2* | 10 | 20 | 59.66278 | 10.62596 | This study |
| Skagerrak, Sweden | *SKA* | 2 | n/a | 58.18000 | 11.47000 | *Laakkonen et al. (2021)* |
| Vestfjorden, Norway | *VES* | 28 | 10 | 67.21000 | −14.30000 | *Addison & Hart (2005)* |
| Kongsfjord, Norway | *NFV* | 10 | 5 | 70.72000 | 29.44000 | This study |
| Porsangerfjorden, Norway | *PSF* | 21 | 2-5 | 70.27948 | 25.29986 | This study |
| Varanger Peninsula, Norway | *VAR* | 12 | intertidal | 70.28330 | 30.99770 | *Laakkonen et al. (2021)* |
| Kola Peninsula, Russia | *KOL* | 16 | sublittoral | 69.1177 | 36.07680 | *Laakkonen et al. (2021)* |
| White Sea, Russia | *WHS* | 1 | n/a | 66.2900 | 33.61000 | *Laakkonen et al. (2021)* |

length of 418bp in SEQUENCER, version 5.0 (Gene Codes; GenBank accession numbers OL451446–OL451529, OL451534–OL451866).

## Polymorphism and population genetic structure

We identified individuals as *Strongylocentrotus pallidus*, or one of the two reproductively isolated cryptic lineages of *S. droebachiensis* (see *Addison & Kim, 2018*) using a combination of maximum likelihood and statistical parsimony. We inferred a phylogenetic tree of unique haplotypes by maximum likelihood using PHYML 3.0 (*Guindon et al., 2010*), with an HKY85 substitution model, gamma distributed rate heterogeneity at sites, and an SPR tree search. Node support for the putative species clusters was estimated using nonparametric bootstrap analysis with 1,000 replicates. To visualize species assignment, we used statistical parsimony implemented in TCS v.1.21 (*Clement, Posada & Crandall, 2000*) and presented using PopART (http://popart.otago.ac.nz; https://github.com/jessicawleigh/popart). Mean genetic distances (K2P, Kimura two-parameter distances; *Kimura, 1980*) within and between lineages were calculated in MEGA (*Kumar et al., 2018*; *Stecher, Tamura & Kumar, 2020*). We calculated genetic diversity for each species following *Addison & Kim (2018)*. Measurements included: nucleotide diversity ($\pi$), number of segregating sites ($S$), number of haplotypes ($H$), and haplotype diversity ($h$) for each sampling location using DNASP v.5.1 (*Librado & Rozas, 2009*). We tested for departures from neutrality based on allelic states or segregating sites with Fu's $F_S$ (*Fu, 1997*) and Tajima's $D$ (*Tajima, 1989*), respectively, using ARLEQUIN (*Excoffier & Lischer, 2010*). For neutral or near-neutral evolving markers such as mtDNA, significantly negative values of these tests can indicate a higher-than-expected number of single mutations ($D$) or haplotypes ($F_S$) which can result from population expansion (*Ramos-Onsins & Rozas, 2002*). While both tests are frequently used to distinguish between models of population growth or no-growth, simulations have observed that Fu's $F_S$ has greater power to detect population growth (*Ramos-Onsins & Rozas, 2002*). Significance was assessed by 10,000 coalescent simulations. To control for the occurrence of false positives due to multiple comparisons, significance of the *p*-values was determined using the Bonferroni correction. To simultaneously visualize both the phylogenetic relationships and the frequency of each haplotype, we constructed separate haplotype networks for each species using statistical parsimony implemented in TCS v.1.21 (*Clement, Posada & Crandall, 2000*) and presented using PopART.

To evaluate the genetic subdivision among populations of each lineage within and between major oceanographic regions, we calculated global $F_{ST}$ and tested for pairwise genetic differences between populations. We conducted analyses of molecular variation (AMOVA) to test for hierarchical genetic structure both within and among the Pacific and Atlantic Oceans (Fig. 1). We also explored *post hoc* hypotheses based on patterns of pairwise $F_{ST}$ to further refine patterns of substructure. Indices of genetic differentiation ($F_{ST}$ and $\Phi$) were calculated using Kimura two-parameter distances (K2P: *Kimura, 1980*) implemented in ARLEQUIN, and significance was assessed using 10,000 permutations of the data with Bonferroni correction for multiple tests.

# RESULTS

We obtained 418 bp *COI* sequences (positions 6415–6832 of *Jacobs et al., 1988*) for 789 individual sea urchins. There were 60 variable sites and a total of 83 unique haplotypes. Based on maximum likelihood and statistical parsimony analyses (Fig. 1C) our results support the presence of three lineages reported in *Addison & Kim (2018)*. We detected three distinct clusters of haplotypes including *S. pallidus* ($n = 156$), and both lineages of *S. droebachiensis* (*S. droebachiensis* E, $n = 148$; *S. droebachiensis* W, $n = 485$) (Table 2). Mean genetic distance among all the sequences (K2P) was 2.40%, while within lineage mean genetic distance ranged from 0.33% to 0.41% (Fig. 1). Pairwise genetic distance was the greatest between *S. pallidus* and *S. drobachiensis* W, while the cryptic lineages within *S. droebachiensis* were 2.73% divergent. For all 3 lineages, the net between group distances (K2P) were 6-28x greater when comparing samples between oceans (*i.e.*, trans-Arctic) than between the east and west coasts of the same ocean (trans-Pacific or trans-Atlantic; Table 3). Overall, haplotype ($h$) and nucleotide ($\pi$) diversity was high in all three lineages, and values ranged from 0.685–0.733 and 0.033–0.0040, respectively (Table 2). Significant negative values of Fu's $F_S$ ($-6.600$ to $-4.400$) suggest a demographic expansion (or purifying selection) in samples of *S. pallidus* and *S. droebachiensis* E from offshore sites (>60 m) in the northwest Atlantic, and *S. droebachiensis* W from OKH and NOM in northeast Pacific (Table 2). Sea urchin populations of all three lineages did not show an excess of alleles in shallow water habitats throughout the northwest Atlantic.

## Biogeography

*Strongylocentrotus pallidus* was distributed in high relative abundance across all three oceanographic regions sampled. Pure populations of *S. pallidus* were detected above the Arctic Circle in both the northwest and northeast Atlantic (Fig. 1), and this species was relatively abundant (13–25%) in mixed aggregations at shallow sites along the coast of Labrador and western Newfoundland. However, *S. pallidus* was rare at all other shallow water sites in Atlantic Canada (six of 236 total samples; 2.5%), including one individual at 14 m in the Bay of Fundy where it was absent at depths >70 m. In contrast, *S. pallidus* was common at deeper sites, making up 28% of the samples collected at 60 m off the coast at Owl's Head, NS (19 of 69), and 69.6% of the samples collected Offshore at depths >65 m on the Scotian Shelf (39 of 56).

The green sea urchin, *S. droebachiensis*, was detected across all three oceanic regions sampled, but there were striking differences in the distribution of the cryptic lineages. *Strongylocentrotus droebachiensis* E was the only green sea urchin found in the northeast Atlantic, where pure populations were sampled throughout Iceland, Norway, and Russia (Fig. 1). In the northwest Atlantic, *S. droebachiensis* E shared a distribution similar to *S. pallidus* where it made up ∼12% of the samples at shallow sites at higher latitudes throughout the Labrador Sea and the Gulf of St Lawrence (20/162). However, this lineage was rare in the shallow coastal samples (<60 m) at lower latitudes, comprising only ∼1% of the individuals sampled along Nova Scotia and the Bay of Fundy (two of 201), and was the only green sea urchin found offshore on the Scotian Shelf. With the exception of the

**Table 2** **Mitochondrial DNA (*COI*) diversity for *Strongylocentrotus* sea urchins from individual sites and within a priori groups.** Number of individuals sequenced (N), number of haplotypes ($H$), number of segregating sites ($S$), nucleotide diversity ($\pi$), haplotype diversity ($h$), and neutrality tests (Tajima's $D$; Fu's $F$). Neutrality tests significantly different from 0 after Bonferroni correction ($P < 0.0056$) are indicated by an asterisk (*).

| Species | Sample Site | Abbr. | Group | N | $H$ | $S$ | | $h$ | $D$ | $F$ |
|---|---|---|---|---|---|---|---|---|---|---|
| *Strongylocentrotus pallidus* | | | | | | | | | | |
| | San Juan Islands, WA | SJI | SJI | 45 | 10 | 10 | 0.0030 (0.0007) | 0.544 (0.085) | −1.324 | −4.43 |
| | Qikiqtarjuaq, NU | QIK | QIK | 5 | 2 | 1 | 0.0010 (0.0006) | 0.400 (0.237) | −0.817 | 0.090 |
| | Labrador Sea: | | LAS | 20 | 4 | 4 | 0.0028 (0.0005) | 0.537 (0.099) | 0.078 | 0.335 |
| | Nain, NL | NAI | | 16 | 3 | 3 | 0.0027 (0.0005) | 0.542 (0.265) | – | – |
| | Makkovik, NL | MAK | | 4 | 2 | 3 | 0.0036 (0.0019) | 0.500 (0.104) | – | – |
| | Atlantic Coast Shallow: | | ACS | 7 | 4 | 6 | 0.0062 (0.0014) | 0.810 (0.123) | 0.254 | 0.354 |
| | Bonne Bay, NL | BBY | | 1 | 1 | 0 | 0 | 0 | – | – |
| | Bear Cove, NS | BCV | | 3 | 2 | 1 | 0.0016 (0.0008) | 0.667 (0.314) | – | – |
| | Duncan Cove, NS | DUN | | 2 | 1 | 0 | 0 (0.0004) | 0 | – | – |
| | Bay of Fundy (shallow) | BFS | | 1 | 1 | 0 | 0 (0.0004) | 0 | – | – |
| | Owl's Head, NS | OWH | OWH | 19 | 7 | 7 | 0.0020 (0.0006) | 0.608 (0.127) | −1.954 | −4.400* |
| | Atlantic Coast Offshore: | | ACO | 39 | 6 | 5 | 0.0008 (0.0003) | 0.327 (0.095) | −1.800 | −4.891* |
| | NS offshore (65m) | 065 | | 7 | 2 | 1 | 0.0007 (0.0005) | 0.286 (0.196) | – | – |
| | NS offshore (72m) | 072 | | 14 | 3 | 2 | 0.0010 (0.0004) | 0.385 (0.149) | – | – |
| | NS offshore (92m) | 092 | | 12 | 4 | 3 | 0.0012 (0.0005) | 0.455 (0.170) | – | – |
| | NS offshore (120m) | 120 | | 6 | 1 | 0 | 0 | 0 | – | – |
| | Porsangerfjorden, NOR | PSF | NOR | 21 | 3 | 2 | 0.0005 (0.0003) | 0.186 (0.110) | −1.514* | −1.920 |
| | **Total** | | | **156** | **20** | **18** | **0.0033 (0.0004)** | **0.685 (0.032)** | **−1.526** | **-12.734*** |
| *Strongylocentrotus droebachiensis* E | | | | | | | | | | |
| | San Juan Islands, WA | SJI | SJI | 5 | 5 | 8 | 0.0100 (0.0016) | 1.000 (0.013) | 0.477 | −1.674 |
| | Labrador Sea: | | LAS | 11 | 3 | 3 | 0.0020 (0.0008) | 0.345 (0.172) | −1.113 | −0.113 |
| | Durban Island, NU | DUR | | 1 | 1 | 0 | 0 | 0 | – | – |
**Table 2** (*continued*)

| Species | Sample Site | Abbr. | Group | N | H | S | | h | D | F |
|---|---|---|---|---|---|---|---|---|---|---|
| | Nain, NL | NAI | | 10 | 2 | 2 | 0.0009 (0.0007) | 0.200 (0.154) | – | – |
| | **Atlantic Coast Shallow** | | ACS | 13 | 5 | 8 | 0.0040 (0.0015) | 0.628 (0.143) | −1.37 | −0.504 |
| | Conception Bay, NL | CBY | | 1 | 1 | 0 | 0 | 0 | – | – |
| | Bonne Bay, NL | BBY | | 2 | 1 | 0 | 0 | 0 | – | – |
| | Harve Saint Pierre, QC | HSP | | 1 | 1 | 0 | 0 | 0 | – | – |
| | Miramichi, NB | MIR | | 5 | 1 | 0 | 0 | 0 | – | – |
| | Jeddore, NS | JED | | 2 | 1 | 0 | 0 | 0 | – | – |
| | Bear Cove, NS | BCV | | 2 | 2 | 5 | 0.0120 (0.0060) | 1.000 (0.500) | – | – |
| | **Owl's Head (65m)** | OWH | OWH | 9 | 7 | 7 | 0.0051 (0.0009) | 0.944 (0.070) | −0.804 | −3.618 |
| | **Atlantic Coast Offshore** | | ACO | 19 | 11 | 10 | 0.0045 (0.0007) | 0.865 (0.071) | −1.154 | −6.600* |
| | NS offshore (65m) | 065 | | 1 | 1 | 0 | 0 | 0 | – | – |
| | NS offshore (72m) | 072 | | 5 | 4 | 6 | 0.0062 (0.0017) | 0.900 (0.161) | – | – |
| | NS offshore (92m) | 092 | | 2 | 2 | 2 | 0.0048 (0.0024) | 1.000 (0.500) | – | – |
| | NS offshore (120m) | 120 | | 9 | 5 | 5 | 0.0033 (0.0010) | 0.806 (0.120) | – | – |
| | Bay of Fundy (deep) | BFD | | 2 | 2 | 1 | 0.0024 (0.0012) | 1.000 (0.500) | – | – |
| | **Hvalfjordur, Iceland** | ICE | ICE | 12 | 4 | 3 | 0.0023 (0.0007) | 0.561 (0.154) | −0.128 | −0.719 |
| | **Skagerrak, Sweden** | SKA | ND2 | 2 | 1 | 1 | 0 | 0 | – | – |
| | **Oslo fjord, Norway** | ND2 | ND2 | 10 | 2 | 2 | 0.0017 (0.0008) | 0.356 (0.159) | 0.019 | 1.532 |
| | **Vestfjorden, Norway** | VES | VES | 28 | 4 | 3 | 0.0016 (0.0002) | 0.587 (0.048) | −0.3387 | −0.6325 |
| | **Kongsfjord, Norway** | NFV | NFV | 10 | 1 | 0 | 0 | 0 | – | – |
| | **Varanger Peninsula, Norway** | VAR | VAR | 12 | 3 | 3 | 0.0021 (0.0007) | 0.621 (0.087) | −0.3785 | 0.4281 |
| | **Kola Peninsula, Russia** | KOL | KOL | 16 | 4 | 2 | 0.0017 (0.0004) | 0.592 (0.122) | 0.5192 | −0.9678 |
| | **White Sea, Russia** | WHS | KOL | 1 | 1 | 0 | 0 | 0 | – | – |
| | **Total** | | | **147** | **28** | **23** | **0.0033 (0.0003)** | **0.715 (0.036)** | **−0.388** | **−0.169** |

| Species | Sample Site | Abbr. | Group | N | *H* | *S* | | *h* | *D* | *F* |
|---|---|---|---|---|---|---|---|---|---|---|
| *Strongylocentrotus droebachiensis* W | | | | | | | | | | |
| | Sea of Okhotsk, Russia | OKH | RUS | 16 | 6 | 5 | 0.0018 (0.0006) | 0.542 (0.147) | −1.692 | −3.693* |
| | San Juan Islands, WA | SJI | SJI | 29 | 8 | 9 | 0.0029 (0.0006) | 0.702 (0.059) | −1.496 | −3.277 |
| | Queen Charlotte Islands: | | QCI | 38 | 6 | 5 | 0.0014 (0.0004) | 0.413 (0.097) | −1.273 | −2.962 |
| | Massett, BC | MAS | | 18 | 4 | 3 | 0.0020 (0.0004) | 0.595 (0.109) | – | – |
| | Langara Island, BC | LAN | | 20 | 3 | 3 | 0.0007 (0.0005) | 0.195 (0.115) | – | – |
| | Kachemak Bay, Alaska | KAC | | 2 | 1 | 0 | 0 | 0 | – | – |
| | Nome, AK | NOM | NOM | 26 | 10 | 9 | 0.0038 (0.0006) | 0.834 (0.054) | −1.08 | −4.832* |
| | Durban Island, NU | DUR | DUR | 18 | 5 | 4 | 0.0022 (0.0005) | 0.641 (0.097) | −0.673 | −1.521 |
| | Nain, NL | NAI | NAI | 39 | 6 | 7 | 0.0021 (0.0006) | 0.437 (0.093) | −1.336 | −1.773 |
| | Makkovik, NL | MAK | MAK | 27 | 5 | 5 | 0.0021 (0.0007) | 0.484 (0.104) | −0.932 | −1.123 |
| | Mid-Atlantic Shallow: | | MAS | 32 | 5 | 4 | 0.0029 (0.0004) | 0.619 (0.084) | 0.5347 | −0.066 |
| | Conception Bay, NL | CBY | | 6 | 3 | 2 | 0.0027 (0.0012) | 0.733 (0.155) | – | – |
| | Bonne Bay, NL | BBY | | 4 | 2 | 3 | 0.0036 (0.0019) | 0.500 (0.265) | – | – |
| | Harve Saint Pierre, QC | HSP | | 6 | 2 | 3 | 0.0038 (0.0006) | 0.533 (0.172) | – | – |
| | Miramichi, NB | MIR | | 16 | 5 | 4 | 0.0028 (0.0006) | 0.667 (0.113) | – | – |
| | Jeddore, NS | JED | JED | 40 | 9 | 9 | 0.0034 (0.0005) | 0.697 (0.007) | −0.959 | −2.96 |
| | Bear Cove, NS | BCV | BCV | 43 | 7 | 5 | 0.0027 (0.0005) | 0.589 (0.082) | −0.047 | −1.749 |
| | Duncan Cove, NS | DUN | DUN | 46 | 4 | 3 | 0.0021 (0.0004) | 0.409 (0.085) | 0.535 | 0.395 |
| | Owl's Head, NS | OWH | OWH | 41 | 4 | 3 | 0.0017 (0.0005) | 0.411 (0.087) | −0.011 | −0.197 |
| | Bay of Fundy: | | BOF | 88 | 4 | 3 | 0.0023 (0.0003) | 0.564 (0.051) | 1.178 | 1.282 |
| | Bay of Fundy (deep) | BFD | | 26 | 3 | 3 | 0.0023 (0.0005) | 0.446 (0.105) | – | – |
| | Bay of Fundy (shallow) | BFS | | 62 | 4 | 3 | 0.0024 (0.0003) | 0.605 (0.054) | – | – |
| | **Total** | | | **485** | **33** | **27** | **0.0040 (0.0001)** | **0.733 (0.013)** | **−0.463** | **−1.729** |

**Notes.**
Totals are shown in bold.

**Table 3 Pairwise genetic distances (K2P) within and between oceanic regions for *Strongylocentrotus pallidus* (*S.p*), *S. droebachiensis* E (*S.d.*E), and *S. droebachiensis* W (*S.d.*W).**

| Ocean Basin | Species | NW Pacific | NE Pacific | NW Atlantic | NE Atlantic |
|---|---|---|---|---|---|
| NW Pacific | *S.p* | – | – | – | – |
| | *S.d.*E | – | – | – | – |
| | *S.d.*W | 0.0018 | 0.0001 | 0.0025 | |
| NE Pacific | *S.p* | – | 0.0005 | 0.0013 | 0.0013 |
| | *S.d.*E | – | 0.0099 | 0.0022 | 0.0028 |
| | *S.d.*W | – | 0.0031 | 0.0022 | |
| NW Atlantic | *S.p* | – | – | 0.0032 | 0.0002 |
| | *S.d.*E | – | – | 0.0046 | 0.0001 |
| | *S.d.*W | – | – | 0.0034 | – |
| NE Atlantic | *S.p* | – | – | – | 0.0030 |
| | *S.d.*E | – | – | – | 0.0024 |
| | *S.d.*W | – | – | – | – |

five individuals from the San Juan Island site reported in *Addison & Hart (2005)*, we failed to detect additional samples of *S. droebachiensis* E throughout the north Pacific.

*Strongylocentrotus droebachiensis* W was distributed throughout the Pacific and coastal samples from Atlantic Canada. With the exception of the low frequency and geographically isolated haplotypes of *S. droebachiensis* E detected at SJI, all other green sea urchin samples collected throughout the Pacific Ocean were identified as *S. droebachiensis* W (Fig. 1). In the northwest Atlantic, *S. droebachiensis* W was detected at all coastal sites (including OWH and BOF) where it comprised an increased proportion of the samples at lower latitudes (Fig. 1). With the exception of the pure population of *S. pallidus* sampled at QIK, samples of sea urchins collected in the Gulf of St. Lawrence and the Labrador Sea consisted of 73.4% *S. droebachiensis* W (116 of 157) compared to 86.6% (258/298) of those sampled along Nova Scotia and the Bay of Fundy at the southern end of its' range. When the deep-water samples in the Bay of Fundy (70–90 m) and Owl's Head (60 m) are removed, the proportion of the *S. droebachiensis* W lineage in the shallow habitat (<30 m) throughout Nova Scotia and the Bay of Fundy increases to 95% (191 of 201).

## Population genetic structure
### *Strongylocentrotus pallidus*
Range-wide genetic structure in *S. pallidus* was primarily driven by differences between the Pacific and Atlantic samples, and an absence of genetic subdivision within the north Atlantic. Statistical parsimony identified a single abundant and geographically widespread genetic variant (50% of all samples) distributed across all three major oceanographic regions. In addition to this shared haplotype, there was some inter-ocean divergence as half of the haplotypes detected in both the Pacific and the shallow water northwest Atlantic were exclusive to those regions (Fig. 2). In contrast, with only two exceptions, all the *S. pallidus* samples from Norway (PSF) shared a single *COI* haplotype. Multi-locus nuclear genotypes have been scored for nine of the 20 individuals (all from ACS) harbouring the three unique northwest Atlantic *S. pallidus* haplotypes (Fig. 2), all of which have been identified as *S.*

*droebachiensis* W (*Addison & Hart, 2005*; *Addison & Kim, 2018*; Burke, Kim & Addison, in prep.) suggesting historic hybridization and introgression (*Addison & Pogson, 2009*). Since these haplotypes have been segregating in *S. droebachiensis* W for many generations, they do not reflect the biology of *S. pallidus* and thus we removed them from subsequent analyses of population genetic structure. Global $F_{ST}$ was high (0.2816; $P < 0.001$), indicating strong and significant variance in the distribution of genetic variation. Pairwise $F_{ST}$ values were high between the Pacific (SJI) and the northwest Atlantic samples from (OWH and ACO) and northeast Atlantic (PSF; Pairwise $F_{ST} = 0.2998$–0.3704, $P < 0.003$; Table 4), suggesting limited dispersal with gene flow across the Arctic basin. In contrast, there was not a significant difference between sample sites within the north Atlantic (Pairwise $F_{ST}$ = 0–0.0689, $P > 0.003$). Hierarchical analysis of molecular variance (AMOVA) based on the *a priori* grouping of populations from each oceanographic region were not significant ($\Phi_{CT} = 0.353$; $P = 0.205$), and we failed to detect evidence of sub-structure based on geography or depth within the Atlantic (Table 5).

*Strongylocentrotus droebachiensis* E

Patterns of genetic subdivision were largely driven by differences between the Pacific and Atlantic basins, and low substructure within Atlantic. We detected seven *COI* haplotypes *S. droebachiensis* E sampled from Norway and Iceland, four of which were also widespread and abundant in the northwest Atlantic (Fig. 3). Two of the three unique haplotypes in the northeast Atlantic were detected in the easternmost Arctic sites in Norway (VAR) and Russia (KOP). While *S. droebachiensis* E was detected throughout the northwest Atlantic, the majority of the individuals were found at deep sites (>60 m) and had rare mtDNA haplotypes. Of the 28 unique haplotypes detected for this lineage, 21 were only detected once, and of those 14 were found in the northwest Atlantic. None of the five *COI* haplotypes genotyped at SJI in the Pacific were found in the Atlantic populations. Global $F_{ST}$ was lower for this lineage compared to the others ($F_{ST} = 0.2052$; $P < 0.001$), indicating a moderate level of genetic subdivision throughout the range. Pairwise comparisons revealed strong divergence between both SJI (Pacific), and ND2 (Norway) from most other sites (Table 6). Genetic subdivision was generally low and not significantly different from zero among most locations across the Atlantic Basin. Hierarchical AMOVA indicated strong regional grouping based on the oceanographic regions within the northeast Atlantic ($\Phi_{CT} = 0.3287$, $P < 0.0001$; Table 5, AMOVA), but we failed to detect significant variation across the north Atlantic or among the *a priori* grouping of sampling sites across oceanographic basins.

*Strongylocentrotus droebachiensis* W

Genetic variation within *S. droebachiensis* W was consistent with previous studies, with strong differences across the Arctic and genetic homogeneity among southern coastal sites in the northwest Atlantic. However, shared halpotypes and generally lower pairwise $F_{ST}$ values (Table 7) among new samples from the Labrador Sea and north Pacific suggest a greater influence of trans-Arctic dispersal. Unique haplotypes were found in both the Pacific and northwest Atlantic, but three high frequency genetic variants were shared throughout both oceans (Fig. 4). Based on both the distribution of haplotypes and pairwise $F_{ST}$ among new

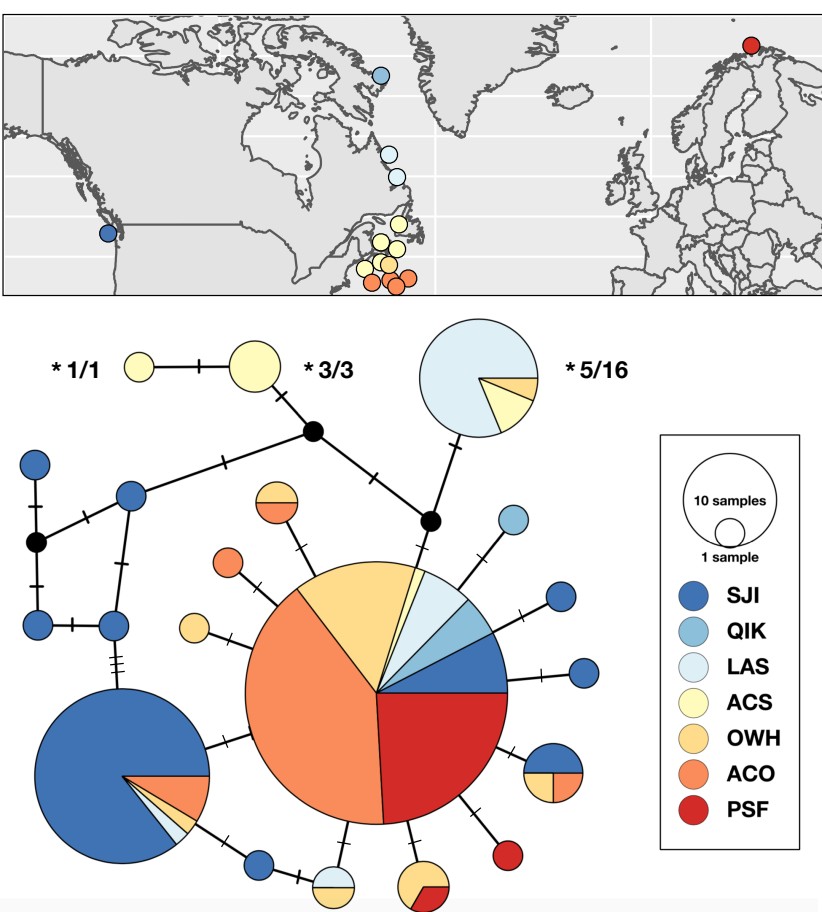

**Figure 2  Sampling locations, haplotype distribution, and TCS haplotype network of COI mtDNA sequences for *Strongylocentrotus pallidus* (*n* = 156).** Asterisks (*) indicate the mtDNA haplotypes removed from analyses of population genetic structure because they were recovered in individuals whose nuclear genomes (SNPs or microsatellites) were characterized as being 100% *S. droebachiensis* W (# tested/# individuals with the haplotype).

**Table 4  Pairwise *$F_{ST}$* values among sampling locations for *Strongylocentrotus pallidus* using mitochondrial DNA (*COI*).** Values of $F_{ST}$ are above the diagonal with significant values in bold, and significance after Bonferroni correction ("+" for $P < 0.0033$, "−" for $P > 0.0033$) is indicated below the diagonal. —indicates no data.

|  | SJI | QIK | LAS | OWH | ACO | PSF |
|---|---|---|---|---|---|---|
| SJI | – | 0.2881 | 0.2023 | **0.2998** | **0.3457** | **0.3704** |
| QIK | – | – | −0.0108 | −0.0214 | 0.0232 | 0.0687 |
| LAS | – | – | – | −0.0211 | −0.0326 | 0.0634 |
| OWH | + | – | – | – | 0.0021 | −0.0054 |
| ACO | + | – | – | – | – | 0.0076 |
| PSF | + | – | – | – | – | – |

**Table 5** Analysis of molecular variance results of mtDNA (*COI*) for three species of *Strongylocentrotus* sea urchins based on *a priori* groupings of sample sites within oceanic regions, and *ad hoc* hypotheses based on analyses of pairwise $F_{ST}$. Significant values ($P < 0.05$) of $\Phi_{CT}$ (variation among groups), $\Phi_{ST}$ (variation within populations), and $\Phi_{SC}$ (variation among populations within groups) are in bold.

| Hypothesis | Grouping | $\Phi_{CT}$ | $\Phi_{ST}$ | $\Phi_{SC}$ | $\Phi_{CT}$ P | $\Phi_{ST}$ P | $\Phi_{SC}$ P |
|---|---|---|---|---|---|---|---|
| ***S. pallidus*** | | | | | | | |
| Among oceanic regions (Pacific/N-WA/NEA) | (SJI) + (QIK, LAS, OWH, ACO) + (PSF) | 0.353 | **0.334** | −0.029 | 0.205 | <0.001 | 0.516 |
| Among oceanic regions, subdivision based on depth within NWA | (SJI) + (QIK, LAS) + (OWH, ACO) + (PSF) | 0.321 | **0.306** | −0.021 | 0.180 | <0.001 | 0.446 |
| Intra-Atlantic (NWA/NEA) | (LAS, QIK, OWH, ACO) + (PSF) | −0.006 | −0.002 | 0.005 | 0.602 | 0.516 | 0.395 |
| ***S. droebachiensis E*** | | | | | | | |
| Among oceanic regions (Pacific/N-WA/NEA) | (SJI) + (LAS,ACS, OWH, ACO) + (ICE, ND2, VES, NFV, VAR, KOL) | 0.091 | **0.237** | **0.161** | 0.086 | <0.001 | <0.001 |
| Intra-Atlantic (NWA/NEA) | (LAS,ACS, OWH, ACO) + (ICE, VES, ND2, NFV, VAR, KOL) | −0.025 | **0.156** | **0.177** | 0.660 | <0.001 | <0.001 |
| Within the NEA only: North Sea, Norwegian Sea, Barents Sea | (ND2) + (ICE, VES) + (NFV, VAR, KOL) | **0.329** | **0.377** | **0.072** | <0.001 | <0.001 | 0.042 |
| ***S. droebachiensis W*** | | | | | | | |
| Among oceanic regions (NWP/NEP/N-WA/NEA) | (OKH) + (SJI, QCI, NOM) + (DUR, NAI, MAK, MAS, JED, BCV, OWH, BOF) | **0.314** | **0.547** | **0.340** | 0.036 | <0.001 | <0.001 |
| Among oceanic regions, north south subdivision in NWA | (OKH) + (SJI, QCI, NOM) + (DUR, NAI, MAK) + (MAS, JED, BCV, OWH, BOF) | **0.497** | **0.539** | **0.084** | 0.001 | <0.001 | <0.001 |
| Grouped by latitude | (NOM, DUR) + (KOH, QCI, NAI, MAK) + (SJI, MAS, JED, BCV, OWH, BOF) | **0.396** | **0.540** | **0.239** | 0.015 | <0.001 | <0.001 |

samples, sites from the Labrador Sea were generally more similar to sites in the north Pacific than to the northwest Atlantic. Global $F_{ST}$ was high (0.4337; $P < 0.0001$), and was largely driven by the differences between the southern samples of the northwest Atlantic (*i.e.*, the Gulf of St. Lawrence, Nova Scotia, and the Bay of Fundy) and those from the Labrador Sea and the north Pacific (Table 7). Consistent with earlier studies (*Addison & Hart, 2004*;
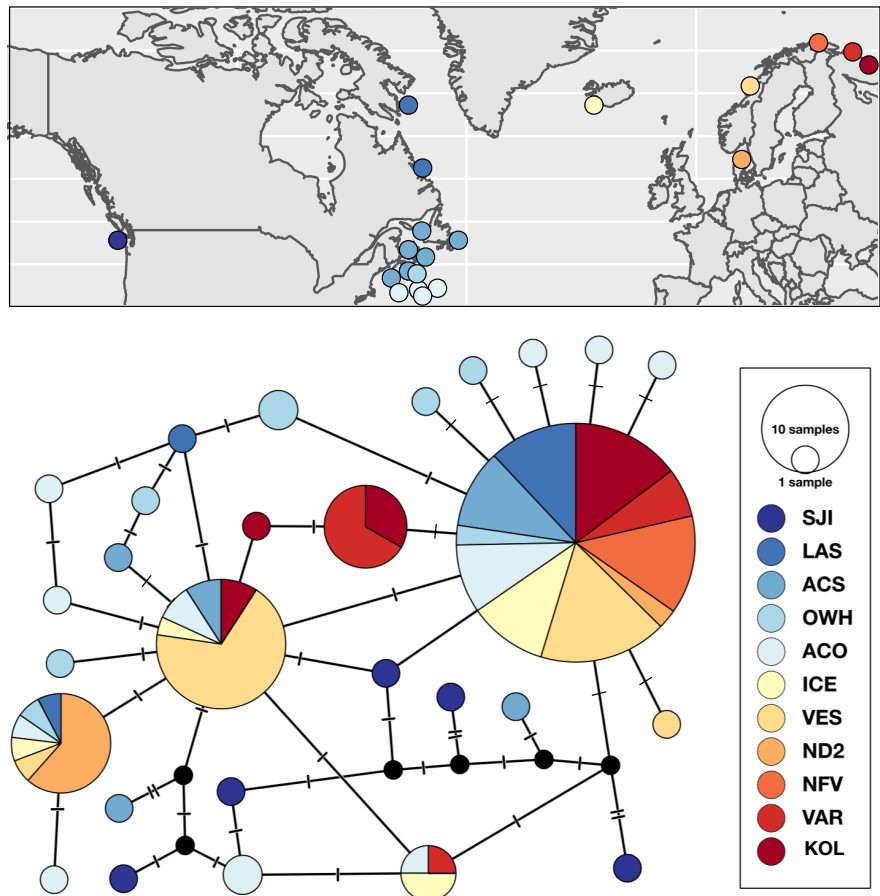

**Figure 3** Sampling locations, haplotype distribution, and TCS haplotype network of *COI* mtDNA sequences for *Strongylocentrotus droebachiensis* E (n = 148).

*Addison & Hart, 2005*) sites throughout the Gulf of St. Lawrence, Nova Scotia, and the Bay of Fundy were genetically homogeneous (pairwise $F_{ST} = 0$ to $0.0400$, $P > 0.10$). There was a striking correlation between latitude and genetic similarity among the northern sites in the Pacific and the Labrador Sea. Pairwise $F_{ST}$ was not significant between NOM (Alaska; 64.487°N) and DUR (Nunavut; 67.038°N), but these sites were different from the next closest sample in each region (Table 7). A similar pattern in the magnitude of the pairwise $F_{ST}$ was observed between OKH (Russia; 59.494°N), QCI (British Columbia; 54.193°N) and both NAI (Labrador; 56.504°N) and MAK (Labrador; 55.102°N). This latitudinal pattern was driven by differences in the identity of the single most abundant haplotype at sample sites within each group. The most frequent haplotype at OKH, QCI, NAI, and MAK (0.69, 0.77, 0.74, and 0.70, respectively), was the second most frequent haplotype at NOM (0.23) and DUR (0.28). Although genetic structure based on the *a priori* grouping of samples in the Pacific and Atlantic was significant ($\Phi_{CT} = 0.3184$; $P = 0.0349$; Table 5), as was our grouping of samples by latitude ($\Phi_{CT} = 0.3955$; $P = 0.0146$), exploration of the results maximized the resolution of geographic subdivision when we included four distinct

Table 6 **Pairwise $F_{ST}$ values among sampling locations for *Strongylocentrotus droebachiensis* E using mitochondrial DNA (*COI*).** Values of $F_{ST}$ are above the diagonal with significant values in bold, and significance after Bonferroni correction (" +" for $P < 0.0009$, "−" for $P > 0.0009$) is indicated below the diagonal. —indicates no data.

|  | SJI | LAB | ACS | OWH | ACO | ICE | ND2 | VES | NFV | VAR | KOL |
|---|---|---|---|---|---|---|---|---|---|---|---|
| SJI | – | **0.4549** | 0.3022 | **0.3175** | 0.2921 | **0.3850** | **0.5238** | **0.5568** | **0.5546** | **0.4795** | **0.5090** |
| LAB | + | – | −0.0091 | −0.0025 | 0.0247 | −0.0161 | 0.4852 | 0.0960 | 0.0393 | 0.2134 | 0.0333 |
| ACS | – | – | – | 0.0227 | 0.0046 | −0.0087 | **0.3529** | 0.0432 | 0.0675 | 0.1691 | 0.047 |
| OWH | + | – | – | – | 0.0216 | 0.0325 | 0.0347 | 0.1089 | 0.1513 | 0.2028 | 0.1111 |
| ACO | – | – | – | – | – | −0.0333 | 0.2122 | 0.0129 | 0.1329 | 0.1863 | 0.0898 |
| ICE | + | – | – | – | – | – | 0.3852 | 0.0190 | 0.1570 | 0.2143 | 0.0667 |
| ND2 | + | – | + | – | – | – | – | **0.4022** | **0.7142** | **0.5895** | **0.5369** |
| VES | + | – | – | – | – | – | + | – | 0.3060 | **0.3500** | 0.1589 |
| NFV | + | – | – | – | – | – | + | – | – | 0.3118 | 0.1183 |
| VAR | + | – | – | – | – | – | + | + | – | – | 0.0277 |
| KOL | + | – | – | – | – | – | + | – | – | – | – |

groups of sea urchins in the east and west Pacific, Labrador Sea, and coastal northwest Atlantic ($\Phi_{CT} = 0.49691$; $P = 0.0013$).

## DISCUSSION

The biogeographic distribution, concordant population genetic structure, and patterns of haplotype sharing among oceanic regions suggest that cycles of vicariance and trans-Arctic gene flow has shaped diversification within circumpolar Strongylocentrotid sea urchins. While there is considerable debate about the competing contributions of both geographic isolation and divergence with gene flow to the process of speciation in the sea (*e.g.*, *Miglietta, Faucci & Santini, 2011*; *Faria, Johannesson & Stankowski, 2021*), our results suggest that isolation across the Arctic Basin has been a driving force of genomic and systematic diversity within the genus. Consistent with earlier studies (*Palumbi & Wilson, 1990*; *Palumbi & Kessing, 1991*; *Addison & Hart, 2004*; *Addison & Hart, 2005*; *Harper, Addison & Hart, 2007*), we detected widespread sharing of identical haplotypes throughout the Pacific and Atlantic populations of *S. pallidus* and *S. droebachiensis* W, and patterns of population genetic subdivision among these regions suggests recent interoceanic exchange. However, there was no evidence of a similar pattern of trans-Arctic dispersal in *S. droebachiensis* E, as we failed to detect haplotypes from this species at additional sample sites throughout the north Pacific.

Following the initial trans-Arctic invasion of the north Atlantic by Pacific ancestors, our results suggest that *Strongylocentrotus droebachiensis* diverged into reproductively isolated cryptic species, one of which remains connected with the pacific (*S. droebachiensis* W) while the other is now endemic to the Atlantic (*S. droebachiensis* E). Although the strong patterns of hierarchical population structure within *S. droebachiensis* W suggests a contribution of latitude to the distribution of genetic variation, our analysis of a putatively neutral mtDNA locus does not display a similar signature of adaptive evolution in response to temperature driven by latitudinal variation reported for populations of Atlantic cod

Addison and Kim (2022), *PeerJ*, DOI 10.7717/peerj.13930

**Table 7** **Pairwise $F_{ST}$ values among sampling locations for *Strongylocentrotus droebachiensis* W using mitochondrial DNA (*COI*).** Values of $F_{ST}$ are above the diagonal with significant values in bold, and significance after Bonferroni correction ("+" for $P < 0.0006$, "−" for $P > 0.0006$) is indicated below the diagonal. A dash (−) indicates no data.

| | OKH | SJI | QCI | NOM | DUR | NAI | MAK | MAS | JED | BCV | DUN | OWH | BOF |
|---|---|---|---|---|---|---|---|---|---|---|---|---|---|
| OKH | − | 0.1712 | −0.0061 | 0.2227 | 0.2437 | −0.0096 | 0.0107 | **0.5553** | **0.5317** | **0.6197** | **0.6771** | **0.7293** | **0.6183** |
| SJI | − | − | **0.2316** | **0.3510** | **0.3614** | **0.2100** | **0.2164** | **0.5932** | **0.5749** | **0.6359** | **0.6877** | **0.7232** | **0.6575** |
| QCI | − | + | − | **0.2217** | 0.2192 | −0.0046 | −0.0006 | **0.5890** | **0.5673** | **0.6470** | **0.6922** | **0.7390** | **0.6287** |
| NOM | − | + | + | − | −0.0105 | **0.1873** | 0.1340 | **0.3333** | **0.3342** | **0.4419** | **0.4836** | **0.5327** | **0.4357** |
| DUR | − | + | − | − | − | 0.1558 | 0.1053 | **0.3756** | **0.3613** | **0.4808** | **0.5376** | **0.6046** | **0.4641** |
| NAI | − | + | − | + | − | − | −0.0205 | **0.5131** | **0.5009** | **0.5826** | **0.6275** | **0.6747** | **0.5735** |
| MAK | − | + | − | − | − | − | − | **0.4852** | **0.4702** | **0.5618** | **0.6134** | **0.6670** | **0.5547** |
| MAS | + | + | + | + | + | + | + | − | −0.0236 | 0.0119 | 0.0112 | 0.0400 | −0.0116 |
| JED | + | + | + | + | + | + | + | − | − | −0.0012 | 0.0038 | 0.0266 | −0.0072 |
| BCV | + | + | + | + | + | + | + | − | − | − | −0.0123 | 0.0018 | 0.0117 |
| DUN | + | + | + | + | + | + | + | − | − | − | − | −0.0106 | 0.0034 |
| OWH | + | + | + | + | + | + | + | − | − | − | − | − | 0.0126 |
| BOF | + | + | + | + | + | + | + | − | − | − | − | − | − |

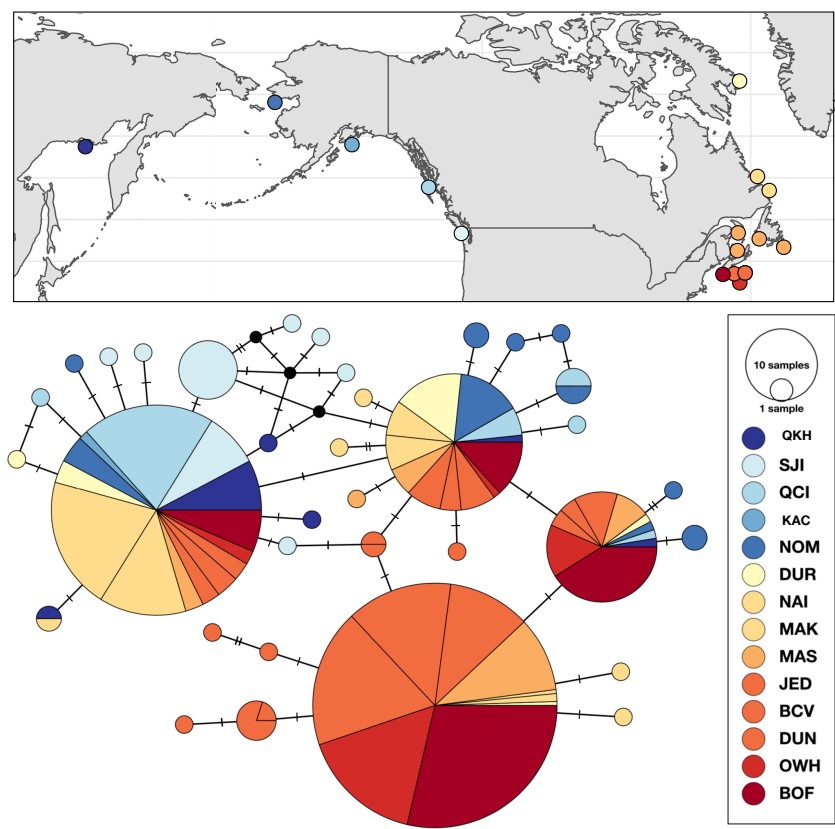

**Figure 4** Sampling locations, haplotype distribution, and TCS haplotype network of *COI* mtDNA sequences for *Strongylocentrotus droebachiensis* W (*n* = 485).

(*Bradbury et al., 2010*) and Atlantic salmon (*Jeffery et al., 2017*). Our results suggest that repeated trans-Arctic gene exchange contributed to the maintenance of species integrity in two species, while vicariance may have contributed to the allopatric speciation of a third species in the north Atlantic.

Like many arctic-boreal marine species, *Strongylocentrotus* sea urchins with a circumarctic distribution have experienced cycles of isolation and invasion throughout the Pleistocene ice ages (*e.g.*, *Laakkonen et al., 2021*). Following the trans-Arctic invasion in the late Pliocene, allopatric populations of Pacific and Atlantic sea urchins diverged throughout the Pleistocene when cycles of glacial advance and retreat (ca 2.4–3.0 Mya) restricted dispersal around the Bering Strait (*Einarsson, Hopkins & Doell, 1967*; *Herman & Hopkins, 1980*; *Maslin et al., 1996*; *Haug et al., 1999*; *Harris, 2005*; *Lisiecki & Raymo, 2005*; *Horikawa et al., 2015*; *Loeza-Quintana et al., 2019*). The presence of multiple private haplotypes in Nova Scotia and New Brunswick suggests that *S. droebachiensis* W persisted in refugia at the southern end of their northwest Atlantic range, but moderate genetic structure and sequence similarity among oceans indicates that Pacific populations subsequently re-invaded the Atlantic during interglacial periods throughout the Pleistocene (*Hewitt, 2003*; *Maggs et al., 2008*). We detected patterns of genetic subdivision and sequence diversity

within *S. pallidus* that are consistent with *S. droebachiensis* W, although the small sample size of the hierarchical analysis (three groups, six sites) suffered from low power and could not approach significance at the 5% threshold (*Fitzpatrick, 2009*). Genetic analysis of new samples collected at high latitudes in both oceans revealed extensive haplotype sharing and lower pairwise $F_{ST}$ values compared to studies conducted at more southern latitudes (*Palumbi & Wilson, 1990*; *Palumbi & Kessing, 1991*; *Addison & Hart, 2004*; *Addison & Hart, 2005*; *Harper, Addison & Hart, 2007*). These results indicate that trans-Arctic connectivity is likely greater than previously reported, and while qualitatively consistent with coalescent analyses (*Laakkonen et al., 2021*) that show the predominant migration vectors track the east flowing currents connecting the Pacific with the northwest Atlantic (*Ledu et al., 2008*), we cannot exclude a hypothesis of back migration from the northwest Atlantic to the Pacific (*see Addison & Hart, 2005*; *Harper, Addison & Hart, 2007*). While Pacific and northwest Atlantic populations of *S. pallidus* and *S. droebachiensis* W experienced periods of vicariance throughout the Pleistocene, late glacial and post-glacial trans-Arctic dispersal continues to maintain the integrity of these species.

Patterns of biogeography and genetic diversity suggest that *S. droebachiensis* E is almost exclusively limited to the Arctic and sub-Arctic in north Atlantic. Additional sampling confirmed that this lineage is the only green sea urchin found in the northeast Atlantic, and that its range in the northwest Atlantic is characterized by a clear shift to deeper habitats at lower latitudes. While *S. droebachiensis* E was present at some shallow sites in the Canadian Arctic, Labrador Sea, and Gulf of St. Lawrence, our sampling efforts at similar latitudes in the Pacific failed to detect additional evidence of this species beyond those reported by *Addison & Hart (2005)*. Since the sites at Haida Gwaii (LAN, MAS), Sea of Okhotsk (OKH), and the Bering Sea (NOM) share similar coastal temperatures and macroalgal assemblages (*Payne et al., 2012*; *Government of Canada, 2014*; *Bringloe & Saunders, 2019*) as those supporting *S. droebachiensis* E in the north Atlantic, failure to detect additional representatives suggests they are not broadly distributed throughout the Pacific. In addition, the five singleton haplotypes of *S. droebachiensis* E reported from the San Juan Islands (*Addison & Hart, 2005*) were not identified in the north Atlantic, indicating very limited (or complete absence) of trans-Arctic gene flow in this species. These results suggest that the presence of *S. droebachiensis* E haplotypes in the Pacific could represent incomplete lineage sorting of ancestral alleles, or possibly low levels of back migration of *S. droebachiensis* E individuals or haplotypes (via introgression into *S. pallidus* or *S. droebachiensis* W; *Addison & Hart, 2005*; *Harper, Addison & Hart, 2007*) during interglacial periods throughout the Pleistocene. While additional analyses of both coastal and deep habitats throughout the Pacific are required before concluding that *S. droebachiensis* E is absent from the Pacific Ocean, our findings suggest this species may have evolved in allopatry following the initial trans-Arctic invasion of *S, droebachiensis* during the late Pliocene. Alternatively, the two lineages of *S. droebachiensis* could have initially diverged in the Pacific prior to invading the Atlantic, followed by a subsequent reduction (or possibly extirpation) *S. droebachiensis* E in the Pacific. At the very least, our study reveals that, unlike *S. pallidus* and *S. droebachiensis* W, Pacific and Atlantic populations of *S. droebachiensis* E continue to diverge in a state of persistent trans-Arctic vicariance.

The repeated trans-Arctic dispersal of *S. pallidus* and *S. droebachiensis* W following the initial period of vicariance suggests that the northwest Atlantic is a zone of secondary contact between all three species. Early studies employing microsatellites (*Addison & Hart, 2005*), nuclear DNA sequences (*Addison & Pogson, 2009*), and single nucleotide polymorphisms (SNPs; *Addison & Kim, 2018*) detected mitochondrial and nuclear discordance in 9 of the 305 (3.0%) individual sea urchins analyzed throughout the Pacific and Atlantic oceans. In these studies, all the hybrid individuals identified were a result of introgression of *S. pallidus* mtDNA into *S. droebachiensis* individuals. For example, *Addison & Kim (2018)* tested for evidence of hybridization using both *COI* sequences and 3,049 nuclear SNPs in a sample of 110 sea urchins collected along a depth gradient off the coast of Nova Scotia. While two *S. droebachiensis* W individuals from shallow sample sites harboured *S. pallidus* mtDNA, the lack of admixture across the nuclear genome of all samples provides evidence against widespread contemporary hybridization and suggests that reproductive isolation is complete. In Addition, patterns of endemism of the introgressed haplotypes in both oceans suggests that historic introgressive hybridization from *S. pallidus* into *S. droebachiensis* W may have occurred independently in Pacific and Atlantic populations. Previous studies have not revealed evidence of hybridization between *S. droebachiensis* E and the other two species. However, extensive analyses of both nuclear and mitochondrial DNA throughout the northwest Atlantic are needed to test the hypothesis that contemporary hybrids form under natural spawning conditions, particularly at sites where all 3 species co-occur (*e.g.*, OWH and NAI).

While trans-Arctic vicariance is the dominant mechanism driving the initial divergence of *S. droebachiensis* E from ancestors in the Pacific, allopatry within the Atlantic has contributed to patterns of divergence in other echinoderms. Beginning in the mid-Pliocene, rapid ocean cooling and the formation of the Labrador current isolated temperate north Atlantic species where warmer mid-Atlantic and Gulf stream waters provided refuge on north American and European coasts (*Berggren & Hollister, 1974*; *Franz, Worley & Merrill, 1981*; *Cronin, 1988*; *Wares, 2001*). Genetic evidence supports this hypothesis in sea stars, where western *Asterias forbesi* and eastern *A. rubens* diverged in allopatry followed by the post-glacial recolonization and sympatry in the northwest Atlantic (*Wares, 2001*; *Wares & Cunningham, 2001*). These species now form a secondary contact zone from Nova Scotia to Cape Cod, and laboratory studies of sperm competition (*Harper & Hart, 2005*), morphology, and genetic surveys of natural populations (*Harper & Hart, 2007*) have identified hybridization and introgression. While patterns of ecological, morphological, and genetic divergence identified within *S. droebachiensis* are qualitatively similar to those for *Asterias*, our results only weakly fit the scenario of post Pliocene divergence of allopatric populations within the north Atlantic. Support for this hypothesis includes evidence of reproductive isolation between the east and west lineages (*Addison & Kim, 2018*), habitat segregation of the eastern lineage in the west, and a signal of range expansion in western samples of *S. droebachiensis* E and the co-distributed population of *S. pallidus*. However, we failed to detect moderate or weak population genetic structure typical of recent trans-Atlantic dispersal (*Young et al., 2002*; *Provan, Wattier & Maggs, 2005*; *Chevolot et al., 2006*; *Jolly et al., 2006*; *Hoarau et al., 2007*; *Souche et al., 2015*; *Andrews et al., 2019*; *Neiva*

*et al., 2020*), and in contrast, we identified more private haplotypes and higher genetic diversity ($h$, $\pi$) in northwest Atlantic samples of both *S. pallidus* and *S. droebachiensis* E. These patterns suggest that sea urchins in the northwest Atlantic have persisted in single or multiple glacial refugia (*Hewitt, 2003*; *Maggs et al., 2008*), and were unlikely to have been extirpated during glacial maxima throughout the Pleistocene. Although repeated cycles of isolation and dispersal between the east and west coasts throughout the Pleistocene may have obscured signals of historic vicariance (*Jesus et al., 2006*; *Maggs et al., 2008*), our results suggest that lineages of *S. droebachiensis* have not been strictly allopatric within the north Atlantic following the initial invasion, and that vicariance within the Atlantic was not the principal driver of speciation within the genus.

Identifying the mechanisms driving speciation in the sea can be challenging because of the difficulty in identifying barriers to gene exchange, or the environmental factors driving adaptation. *Addison & Kim (2018)* suggest that tolerance of seasonally lower salinity may contribute to the ecological segregation of the *Strongylocentrotus* lineages in the southern part of their western Atlantic range (*e.g.*, along the coast of Nova Scotia). In this study, we identified contrasting patterns of geographic distribution and habitat segregation that suggest increased water temperatures in the northwest may contribute to the near complete absence of *S. droebachiensis* E from shallow sites dominated by *S. droebachiensis* W. In the northwest Atlantic, larvae of *S. droebachiensis* grow rapidly at 14 °C (*Hart & Scheibling, 1988*), and in the Pacific and northwest Atlantic both larvae and adults can withstand temperatures up to 19.7 and 22 °C, respectively (*Scheibling & Stephenson, 1984*; *Pearce et al., 2005*). Like other species with planktonic dispersing larvae, *S. droebachiensis* exhibits large regional and interannual fluctuations in recruitment (*e.g.*, *Raymond & Scheibling, 1987*; *Scheibling & Raymond, 1990*; *Scheibllng, 1996*), but is known to settle along the coast of Nova Scotia in July when water temperature can exceed 14 °C (*Balch & Scheibling, 2000*). In August and September, the nearshore water temperatures along the coast of Nova Scotia regularly reach 20 °C (*Scheibling, Feehan & Lauzon-Guay, 2013*). In contrast, water temperatures along the Norwegian coast are comparatively cooler (*Danielssen, Svendsen & Ostrowski, 1996*; *Ibrahim et al., 2014*), and green sea urchins experience recruitment failure in kelp beds at southern latitudes when temperatures exceed 10 °C (*Fagerli, Norderhaug & Christie, 2013*; *Rinde et al., 2014*; *Nyhagen, Christie & Norderhaug, 2018*). By limiting sea urchin recruitment, ocean warming is thought to be a driver of ecological change in Norway, as the southern boundary (65°70′N; *Fagerli, Norderhaug & Christie, 2013*) between kelp-dominated habitat and overgrazed urchin barren grounds continues to shift northward with corresponding increases in water temperature (*Rinde et al., 2014*).

Differences in thermal tolerance among lineages of *S. droebachiensis* may explain the habitat segregation we observed in the northwest Atlantic. The extreme rarity of *S. droebachiensis* E in the shallow habitat along the coast of Nova Scotia could be driven by seasonally warmer water temperatures resulting in recruitment failure, post-settlement mortality, or mortality of juveniles or adults. While summer ocean temperatures along the coast of Nova Scotia are impacted by the Gulf Stream and storm activity (*Scheibling, Feehan & Lauzon-Guay, 2013*), lower water temperatures in the Gulf of St Lawrence and coastal Newfoundland and Labrador are moderated by the cool south flowing Labrador

Current. The increased abundance of *S. droebachiensis* E (and *S. pallidus*) at depths <15 m throughout this part of the range (*i.e.*, north of Nova Scotia) may be explained by seasonal temperatures at or below the 10 °C threshold observed in the northeast Atlantic. The influence of temperature on the distribution of *S. droebachiensis* E is supported by both the decrease in recruitment success along the coast of Norway (*Fagerli, Norderhaug & Christie, 2013*) and the shifting population dynamics of green sea urchins in Oslofjord along the southern coast of Norway. In a response to increased sea surface temperatures (SST), *Nyhagen, Christie & Norderhaug (2018)* demonstrated a significant shift in population density from 10–15 m to cooler water at 20 m, and a reduction in both sea urchin size and recruitment success in 1979 and 1992 compared to 2013. Additionally, while sea urchins are present throughout southern Norway, abundant populations typically only occur at depths of 20 m or greater (*e.g.*, site ND2 in this study; *Norderhaug et al. (2016)*). Although changes in coastal SST indicate a rapid warming trend in both the northwest Atlantic (∼1.0 °C per decade) and the margins of Norwegian and North Seas (between ∼0.3 and 0.7 °C per decade), particularly during the planktonic dispersal and settlement of sea urchins from late spring to autumn (*Lima & Wethey, 2012*), the samples analysed in our study were collected over a relatively short time scale (1999–2015) and are unlikely to have captured ongoing changes in sea urchin distribution in response to increasing SST (*e.g.*, *Hobday & Pecl, 2014*). Though we suggest that temperature may be an important factor in defining the range of *S. droebachiensis* E, comparative analyses of the thermal tolerance of larvae, juveniles and adults of both species are required to test this hypothesis.

The evolution of gamete recognition molecules has long been viewed as an important driver of speciation in marine invertebrates (*Vacquier, 1998*; *Palumbi, 2009*; *Lessios, 2011*; *Vacquier & Swanson, 2011*). Interspecific sperm competition in the plankton is mediated by a variety of proteins and carbohydrates (sulfated polysaccharides) coating the sperm and eggs (*Biermann et al., 2004*), and positive selection detected at sperm Bindin (*e.g.*, *Biermann, 1998*) correlates with the strength of reproductive isolation between species (*Zigler et al., 2005*). *Palumbi & Lessios (2005)* showed that, in addition to a steady accumulation of genome divergence over time, the rate of speciation in sea urchins also depends on the rate of evolution of gamete recognition proteins. In their study, *Palumbi & Lessios (2005)* surveyed species in eight genera and showed that the presence of sympatric species was common in genera with rapid evolution of sperm Bindin. Since studies of Bindin evolution within *Strongylocentrotus* only included samples of *S. droebachiensis* from the Pacific (*Biermann, 1998*; *Pujolar & Pogson, 2011*), it is difficult to assess patterns of positive selection and sequence divergence between the cryptic species of *S. droebachiensis*. However, *Marks et al. (2008)* detected 1.5% sequence divergence at sperm Bindin between samples *S. droebachiensis* from Norway, the northwest Atlantic, and northeast Pacific, and based on the conclusions of our study we suggest that this difference represents interspecific divergence. In a series of heterospecific and conspecific crosses between *S. pallidus* and *S. droebachiensis* from the Pacific and *S. droebachiensis* from Norway, *Biermann & Marks (2000)* demonstrated strong asymmetry in fertilization compatibility among allopatric populations. Our data suggests that the allopatric populations studied by *Biermann & Marks (2000)* represent distinct species, where eggs of *S. droebachiensis* E cannot be

fertilized by sperm from *S. pallidus,* and we interpret their results as a test for reproductive isolation among the species. Consistent with *Strathmann (1981)*, eggs of *S. droebachiensis* W were receptive to sperm from both *S. pallidus* and *S. droebachiensis* E, but sperm from *S. droebachiensis* W either failed (*S. pallidus*) or had very low (*S. droebachiensis* E) fertilization rates in heterospecific crosses. Similarly, eggs of *S. droebachiensis* E could not be fertilized by *S. pallidus* sperm, and their sperm successfully fertilized eggs of *S. droebachiensis* W but were not tested with eggs of *S. pallidus*. In addition to ecological segregation, patterns of sperm Bindin evolution and asymmetry in reproductive compatibility may contribute to the maintenance of species boundaries in sympatry in the northwest Atlantic. Detailed studies of gamete recognition molecules and sperm competition among all three species will help to further understand mechanisms driving evolution within the genera.

## CONCLUSION

While previous work identified a reproductively isolated cryptic lineage of *S. droebachiensis* in the northwest Atlantic (*Addison & Kim, 2018*), this study supports the hypothesis that these species formed as a result of vicariant speciation driven by trans-Arctic isolation. Our results show widespread sharing of *S. pallidus* and *S. droebachiensis* W haplotype variants throughout the north Pacific and north Atlantic Oceans, but that *S. droebachiensis* E is largely restricted to the north Atlantic. We detected low genetic subdivision between *S. droebachiensis* W from the north Pacific and the Labrador Sea, suggesting widespread trans-Arctic gene flow in this species. There was weaker evidence of trans-Arctic dispersal in *S. pallidus*, which could possibly be an artefact of poor sampling of this species in the Pacific. Our analyses of biogeography and *COI* sequence diversity suggests that following allopatric speciation during the Pliocene or early Pleistocene, these species established a zone of secondary contact in the northwest Atlantic and the Labrador Sea. In the northwest Atlantic, we identified sites along the coast of Labrador (NAI) and Nova Scotia (OWH) where all three species of *Strongylocentrotus* are abundant, providing a natural laboratory for studying the ecological and molecular aspects driving the evolution of barriers to gene exchange. From a biogeographic perspective, understanding the mechanisms shaping the distribution of the *S. droebachiensis* species throughout the north Atlantic requires experiments to determine the physiological limits of both. We observed patterns of ecological segregation among the species that suggest temperature may play a role in habitat selection, particularly in the warmer water along the coast of Nova Scotia. In addition, while *Addison & Kim (2018)* provided evidence of reproductive isolation among species collected from Nova Scotia, a wider study aimed at detecting hybridization and introgression at nuclear loci is required to characterize the extent of reproductive isolation across a broader range of habitats. Viewed with a species-specific lens, both rapid sequence divergence at sperm Bindin (*Marks et al., 2008*) and the accumulation of interspecific gamete incompatibility (*Biermann & Marks, 2000*) between *S. droebachiensis* W and *S. droebachiensis* E suggests a potential role of reinforcement selection for pre-zygotic isolation (*Coyne & Orr, 2004*) following secondary contact. By characterizing the extent of reproductive isolation, both laboratory studies of sperm competition and interspecific

fertilization combined with analyses of molecular evolution at gamete recognition loci will help to identify mechanisms that drive barriers to gene exchange in natural populations.

## ACKNOWLEDGEMENTS

We thank those who collected samples for us: Dr. Gary Saunders and Dr. Trevor Bringloe (Queen Charlotte Islands and Nome, Alaska); Taylor Burke (Bay of Fundy), and Dr. Marc Anglès d'Auriac (tissue from Porsangerfjorden Norway, and DNA extracts from Oslo fjord and Kongsfjord, Norway). We thank the Hunters and Trappers Associations of Nattivak and Amaruq (Baffin Island) and the Nunatsiavut Government Research Advisory Committee (NL) for permission to sample traditional Inuit territory. We also thank Taylor Burke and Kate Gallant for performing DNA extractions and sequencing the Bay of Fundy samples.

### Funding

This work was supported by the Natural Sciences and Engineering Research Council (Canada), the New Brunswick Innovation Foundation, and the University of New Brunswick. The funders had no role in study design, data collection and analysis, decision to publish, or preparation of the manuscript.

### Grant Disclosures

The following grant information was disclosed by the authors:
Natural Sciences and Engineering Research Council (Canada).
New Brunswick Innovation Foundation, and the University of New Brunswick.

### Competing Interests

The authors declare there are no competing interests.

### Author Contributions

- Jason A. Addison conceived and designed the experiments, performed the experiments, analyzed the data, prepared figures and/or tables, authored or reviewed drafts of the article, and approved the final draft.
- Jinhong Kim performed the experiments, analyzed the data, authored or reviewed drafts of the article, and approved the final draft.

### Field Study Permissions

The following information was supplied relating to field study approvals (i.e., approving body and any reference numbers):

Sea urchins were collected under a Scientific and/or Educational license (Section 52) approved by the Department of Fisheries and Oceans Canada, and the State of Alaska Department of Fish and Game.

## Data Availability

The partial COX1 sequences are available at GenBank: OL451446–OL451529 and OL451534–OL451866.

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
