# Peer review of "Trans-Arctic vicariance in Strongylocentrotus sea urchins"

_PeerJ, doi:10.7717/peerj.13930_

## Round 0.1 · original submission · Major Revisions

I now have reviews back from two expert referees, and as you will see below, while both are supportive of the work and its value to the field, they are divergent in their opinions regarding your manusscript as submitted. One offers relative minor suggestions for revision and is in favor of the manuscript being published, while the other recommends that the current submission be rejected but could be reconsidered following revision. The critical referee objects to exclusion of the Pacific S. droebanchiensis and feels they would fundamentally change the interpretations of the work, because the core conclusions of the study are incompatible with these individuals. Clearly these concerns need to be addressed in a revision, but I believe the authors are capable of responding and so I find myself falling between these referees and am recommending major revisions to the current submission.

Both referees offer valuable constructive feedback on where they feel the manuscript can be improved. If you decide to undertake revisions, please ensure that all review comments are addressed in a rebuttal letter that outlines how you have addressed each comment. Any edits or clarifications mentioned in the rebuttal letter should also be inserted into the revised manuscript where appropriate. It is a common mistake to address reviewer questions in the rebuttal letter but not in the revised manuscript. If a reviewer raised a question, then your readers will probably have the same question so you should ensure that the manuscript can stand alone without the rebuttal letter. Directions on how to prepare a rebuttal letter can be found at: https://peerj.com/benefits/academic-rebuttal-letters/ and please do not hesitate to reach out to me if you need additional guidance.

I look forward to seeing your revised manuscript.

Reviewer 1 ·

Basic reporting

Addison and Kim provide a complementary study to the previous finding of Addison and Kim 2018. They extend the existing dataset on three species of *Strongylocentrotus* sea urchins with mtCOI data encompassing new sampling sites. They confirm the existence of the two lineages of *S. drobachiensis* (w and e) and show their distribution is associated to depth in the North-West Atlantic. They additionally find evidence for trans-Arctic migration in *S. pallidus* and *S. droebchiensis w*.

Overall, I found the manuscript clearly written and the analyses sound.
I do not have any major reservations on the manuscript. You will find some minor comments below.

Experimental design

The methods are adapted to the dataset used.
Only a small comment on the methods (see additional comments).

Validity of the findings

Newly produced raw data is submitted to GenBank and accessible for now as suplemental files.

Additional comments

## small comments

L136: You use here "potential barriers to dispersal", could you provide maybe in the discussion references having evidence for the *a priori* barriers to dispersal you were expecting?

L198-206: Please detail which estimator of Fst is used.

L327: "Negative pairwise FST values were computed"
Given negative Fst values have no biological meaning, it would be better to only talk about non-significant Fst values.

L375-377: Could you explicitly state the argument excluding the hypothesis of the opposite re-invasion? i.e. the Atlantic populations re-invaded northern Pacific regions.

L425: "vicariance within the Atlantic was an unlikely mechanism of speciation", the sole mechanism of speciation? The wording makes me think you would completely exclude some influence of vicariance in the speciation process here, is it what you imply?

## typos

L13: "and" is in italics

L127: "analyses [of] S. droebachiensis"

L283: missing opening parenthesis before OWH

L286, L309: "sample sites"

L317: Missing word after trans-Arctic?

L326-327: "similarity between in the northern sites in the Pacific [...]", something wrong here I think.

L362: "driven [by] latitudinal [...]"

L400: "provided refuge [on?] north American"

L408: "within in"

L492: "may contribute [to] the maintenance"

Reviewer 2 ·

Basic reporting

Review 67773 PeerJ

Decision: Reject, but encourage a resubmission.

General comments: In this paper, Addison and Kim present an expanded global COI dataset of Strongylocentrotus sea urchins, and use this data to investigate patterns of vicariance across the Pacific and Atlantic basins, commenting on intraspecific and interspecific patterns. They suggest trans-Arctic vicariance produced the S. droebachiensis e, endemic to the Atlantic, while S. droebachiensis and S. pallidus are likely more recent migrants from the Pacific into the Atlantic.
In general, the manuscript is very well written, with clear logical progression of ideas, and appropriately prescribed analyses. I enjoyed reading this. I do however, have several concerns which I hope the authors can defend or address. These are listed in order of importance (high to low):

1. The authors systematically ignore the Pacific S. droebanchiensis e mitochondrial sequences in their manuscript, and thus do not address the implications of these individuals for their interpretations. These cannot be conveniently ignored. This aspect tipped my assessment from major revision to rejection (though I encourage a resubmission), as the Pacific presence of S. droebanchiensis e fundamentally changes interpretations. I don’t see how their core conclusions are compatible with these individuals.
2. Figs. 4-6 are impossible to interpret without a map showing the distribution of haplotypes. This added aspect would also make the Pacific S. droebanchiensis e more evident to the reader and harder to ignore. I also found it hard to evaluate results without the geographic context presented as figures.
3. Predictions need refining (see below), particularly as they related to S. droebanchiensis e. Why does the presence of this species broadly vs non-broadly in the Pacific tip the interpretations from speciation in the Pacific towards the Atlantic? Timeframes of predictions/interpretations are also not always clear (Miocene vs Pliocene vs Pleistocene). I also don’t fully understand how S. pallidus can be used to ground truth predictions/results in S. droebanchiensis e (shouldn’t they speak for themselves?).
4. The authors forward conclusions on temperature profiles driving ecological divergence, along with aspects of reproductive isolation, neither of which are assessed here. While fine for the discussion, I do not view these are core conclusions of the paper; the conclusions pertaining to biogeography should be tightened up.
5. There are further contradictory statements (like S. droebanchiensis e present or not present in Northwest Atlantic shallow waters) that need correcting.
6. The authors acknowledge the potential that widespread hybridizations are a possibility, and indeed detect some cases which are removed from their datasets. This is an overall concern, one the authors should address in a clear and critical manner. I think they can provide more insight on future directions that move this topic away from mitochondrial datasets and coarse resolution nuclear markers. This seems inevitable as genomic data becomes more accessible.
7. I was missing some information on how urchins disperse, sperm is mentioned…perhaps a statement in the introduction is needed. How can these species move such global distances? Are migration rates typically high or low?

Specific comments:

Regarding the predictions at lines 129-130, the authors state that if both S. pallidus and S. droebanchiensis e experienced parallel trans-Arctic or post-glacial trans-Atlantic dispersal, then population subdivision should be concordant. But the authors (appear to) assume these events occur at the same time; these migrations may have occurred in both species, but if during differing interstitials, would lead to different degrees of subdivision. Can the authors elaborate a bit more on their prediction here and whether that assumption is a valid one? Equally, the authors attribute conflicting patterns to Arctic and north Atlantic barriers, however, discordant patterns could equally be driven by dispersal along different time frames.

While I commend the authors on having clearly stated predictions, I’m just not sure I’m buying them, as there is an element of “randomness” or uncertainty in terms of when these events occur that break the links in the hypotheses and predictions outlined here.

Perhaps the arguments here are pedantic if the authors are using subdivision as a proxy for the “strength” of the trans-oceanic barriers discussed. But again, do we interpret population subdivision as driven by the strength of the barriers, or the randomness of events combined with low dispersal rates at global distances?

Also regarding predictions, how do the authors account for the presence of S. droebanchiensis e in the Pacific as it relates to trans-Arctic vicariance? The authors state they predict, if true, S. droebanchiensis e will not be broadly distributed in the Pacific? How do the few records fit into that prediction then (introductions, back migration?)? What if both species used to be common in the Pacific, but one is now rare due to localized extinctions during glaciation? What is the most parsimonious answer that leads to the presence of S. droebanchiensis e in both oceans?

At 275-279, the authors indicate some S. pallidus haplotypes were removed from analysis due to introgression with S. droebanchiensis w. While clearly the correct action to take, I wonder if the authors have a sense of how widespread these historical hybridization dynamics are in their dataset. As the entire analysis relies on mitochondrial signal, it’s possible there are (many) other non-detected cases where organellar capture has masked true history and conflated species (especially in the more closely related S. droebanchiensis species). This has the potential to significantly muddle interpretations (regardless of whether the events are ancient or contemporary). This is defended somewhat in the discussion, but the authors may want to expand a bit to distill doubts. For instance, a lack of widespread hybridization is supported by the lack of admixture across nuclear genomes (lines 384), but what genomes are the authors talking about? Context is missing.

These sorts of events speak to an increasing need to move away from organellar datasets towards high fidelity, genomic scale datasets of nuclear variants. This also speaks to the difficulties in assessing gene flow using organellar markers; since the signal is uniparentally inherited, flow is constricted and not necessarily reflective of actual migration rates. I think this can generate considerably noise in the datasets (for instance, the authors speak of Nome and Nunavut having highly similar populations of S. droebanchiensis w, while adjacent populations in both regions were different). Can the authors defend their study from this perspective?

More specific comments:

Abstract: could be more clearly stated the analysis was done on all three species.

117-118: why are introgression or incomplete lineage sorting the only two possibilities forwarded here, are the authors suggesting that S. droebanchiensis e is not truly present in the shallow waters of the Northwest Atlantic? Since this species is present in shallow waters of Northeast Atlantic, does this not point to true presence in shallow waters of northwest Atlantic, where it is perhaps competitively excluded (mostly) by S. droebanchiensis w? Not pointing to one hypothesis over another, just curious why the authors limited their interpretations.

178: node not nodal (or at least I stumbled on that word).

187-189: This may be more for my own comprehension, but I have to admit I’m struggling to understand how significance can be assigned to these values with bootstrap replicates. What is the neutral state of comparison for these tests, aren’t departures considered from 0? I’m not clear on how critical thresholds are established for this metric, as the text simply states this was assessed with 10,000 bootstrap replicates. Would this not simply reflect the consistency of the result, bearing the signatures of the datasets (not random or null expectations?). Or (as I suspect), is the bootstrapping performed on the entire dataset, such that departures are predicated on the idea that the entire dataset represents a null state? Assuming the thresholds are reflective of the dataset, does this not simply identify outliers?

202: here and elsewhere fig citation is italicized, which doesn’t seem necessary

317: sentence appears to be incomplete

327: “genetic similarity between in the northern sites”; remove “in”?

369: glacial coverage was not at its maximum at the start of the Pleistocene, but rather quite late during the last 800 ka, as the amplitudes and timeframes of the glaciation events widened. Consider revising this statement. To be specific, smaller glaciations every 41 ka occurred earlier in the Pleistocene before switching to larger 100 ka cycles. The classic proxy for this are graphs showing fluctuations in heavy and light oxygen isotopes (e.g. Lisiecki and Raymo [2005]). I wonder why dispersal across the Arctic is unlikely during the Pleistocene, as presumably this was possible during interglacials? This is stated a few sentences later.

374: Not all readers understand the provinces you are referring too, here and elsewhere you’ll need to add further context (Canada, northwest Atlantic, ect).

393: Why does natural selection play a role here? It is not enough to test for widespread hybridizations?

400: refuge “for” north?…a word is missing

433-444: The authors suggest water temperatures in Norway are cooler than in nova scotia, however, Norway covers a lot of latitude. Bergen (south Norway) is actually quite comparable to Lunenberg (NS), whereas tromso (north) has a cooler annual profile. So I think the statement needs to be more specific, with proper context for the interpretations of the authors to hold true. If temperature is a limiting factor, how do the authors explain the presence of S. droebanchiensis e near the coastlines of Sweden and Olso, Norway where profiles are comparatively warmer?
Also, what is the timeframe for the accumulation of the genetic data analysed? Is the exclusion of S. droebachiensis e from northwest Atlantic shallow waters an ongoing process, one that was captured in earlier data and potentially actualized in more recent datasets? In the datasets as presented, S. droebachiensis e is not absent from shallow habitat in Nova Scotia, just rare.

483: While it is fine to discuss knowledge on reproductively isolation between the species, I fail to see how the current study contributes to this body of knowledge. As the authors stated, unless widespread hybridization is explicitly tested for in natural populations, the patterns in mitochondrial divergence could be muddled by organellar capture at different time periods. The history as told by the mitochondrial sequences is a proxy for the history of the species, but the manuscript is in danger of conflating those two concepts.

501-502: How can the authors state S. droebachiensis e is restricted to the Atlantic when they have individuals sampled from the Pacific? The significance of those individuals is routinely brushed aside.

506: Is the timeframe for interpretations not within the Pliocene? >5.3 Ma seems like too early of a timeframe to interpret these patterns? Could the initial migration and speciation across the Arctic not have equally taken place during the Pliocene or even the Pleistocene?

508: here, again, the co-occurrence of all species in the pacific is ignored.

510: for the conclusions, I’d suggest the authors steer away from speculation on ecological drivers since they did not explicitly investigate that here. At best, this can be reworded into a future directions statement, but should not read as a core conclusion. Same goes for sperm Bindin.

There are a lot of figures, which can be combined to increase integration and efficiency in comprehending patterns. In particular, Figs 1-3 can be combined into a multi-panel figure since they all display patterns across the species investigated (global one at the top, then Atlantic Canada and network as two panels below).

Figs. 4-6 are impossible to understand without consulting Fig. 1 and making sense of the acronyms. I suggest providing maps to accompany these figures to show how the haplotypes are distributed (essentially replicating figure 1, but displaying intraspecific rather than interspecific variation). These would be two panel figures, as the networks can still be included.

Table 1: since Tajima’s D and Fu’s F essentially measure the same thing (excess of alleles or lack thereof) can the authors comment on the inconsistent significance of results across the two metrics?

Also, there are a lot of table (9 total), can these be combined? (for instance, tables 2-4 display the same information but for the different species). Typically I’d suggest putting some in the supplemental, but they all contribute important information to the manuscript. They just need to be integrated more efficiently.

Experimental design

See above

Validity of the findings

See above, some fundamental flaws in interpretation.

---

## Round 0.2 · Minor Revisions

Thank you for the careful revision of your manuscript. Both referees are satisfied with the revised version and just have some minor comments for you to consider before moving the manuscript forward into production. I do not expect to send it out for further review, and leave it to you as to how you wish to incorporate this feedback.

Reviewer 1 ·

Basic reporting

I have reviewed the modified manuscript and responses to our comments from the first round and am satisfied by the modifications and answers.

I only have a few comments left on the manuscript.

A thought I should have caught on the first round, but which was clarified by the presence of maps in the figures: You should to discuss the hypothesis of introduction events for the presence of S drobachiensis e haplotypes in the Pacific.

L21: mostly restricted [to]

L65: evolution [of] the genus

L121: that [is] abundant

L314: samples from [OWH and ACO]

Experimental design

Nothing more to add

Validity of the findings

Nothing more to add

Additional comments

Nothing more to add

Reviewer 2 ·

Basic reporting

From my initial review: "General comments: In this paper, Addison and Kim present an expanded global COI dataset of Strongylocentrotus sea urchins, and use this data to investigate patterns of vicariance across the Pacific and Atlantic basins, commenting on intraspecific and interspecific patterns. They suggest trans-Arctic vicariance produced the S. droebachiensis e, endemic to the Atlantic, while S. droebachiensis and S. pallidus are likely more recent migrants from the Pacific into the Atlantic."

I also expressed several concerns that led me to reject the paper, namely, how they reconcile the Pacific S. droebanchiensis e haplotypes with their conclusion this species is endemic to the Atlantic, difficult to interpret figures, among others...

The manuscript is improved considerably since the initial submission. The authors provided solid responses in their rebuttals documents, and their efforts are reflected in a manuscript that is easier to comprehend, and does not overreach with their conclusions. I have nothing major to add. Great work revising what were, in retrospect, quite extensive comments.

Just minor comments I jotted down while reading:

98-103: You discuss divergences, please provide the marker so there is context for interpreting these distances. I assume these all refer to coxI

117: "that is abundant"

Experimental design

Good

Validity of the findings

Good

Additional comments

None

---

## Round 0.3 · accepted · Accept

Happy to move this forward into production now.